# The effect of ergometer cycling and visual foraging on brain function: A pilot study

Tamara S. Dkaidek[1], Amelia Dingley[2], Justin Parsler[3], David P. Broadbent[4], Andre J. Szameitat[1], Daniel T. Bishop[1]*

1 Centre for Cognitive and Clinical Neuroscience, Brunel University London, Uxbridge, London, United Kingdom, 2 Institute of Science and Technology, Brunel University London, Uxbridge, London, United Kingdom, 3 Digital Art - Games, Institute of Art and Humanities, Brunel University London, Uxbridge, London, United Kingdom, 4 Centre for Sport Research, Institute for Physical Activity and Nutrition, Deakin University, Burwood, Australia

* daniel.bishop@brunel.ac.uk

## Abstract

Dual-task training comprising cognitive and physical components may enhance cognitive function, and increased prefrontal cortex activation may underpin these improvements. The aim of this pilot study was to examine the effects of cycling and visual foraging on executive function (EF). Twenty-seven participants (mean age $25.44 \pm 4.31$ years) completed four lab-based sessions, one in which their aerobic capacity ($VO_{2max}$) and baseline EF scores assessed were determined, and three randomized experimental conditions: ergometer cycling (EC), visual foraging (VF) and both combined (EC+VF). Participants' EF performance was assessed at baseline, and pre-and post- intervention using the 2-Back task (working memory), the Flanker Task (inhibitory control), and the Wisconsin Card Sorting Task (WCST; task switching). Functional near-infrared spectroscopy (fNIRS) and eye-tracking data were collected throughout each condition. Affective state was assessed via the Affect Grid. Repeated measures ANCOVAs, incorporating baseline EF task scores as covariates, revealed condition x time x covariate interactions for the Flanker task only; task performance of participants with poorer baseline scores improved more profoundly in the EC condition. Subjective arousal and prefrontal cortex (PFC) activation were higher in both cycling conditions relative to VF; hence, ergometer cycling, rather than visual foraging, might be the more impactful intervention in these regards. However, these elevations were not associated with EF enhancements; near-ceiling effects in EF task performance may explain this. The EC condition elicited greater energetic investment than the EC+VF condition; possibly because the secondary VF task distracted from the cycling exercise. PFC activation was only correlated with gaze fixations during the EC+VF condition, potentially reflecting concurrent increases in supply of, and demand for, oxygen during the combined condition.

**Data availability statement:** All relevant data are within the manuscript and its Supporting Information files.

**Funding:** The author(s) received no specific funding for this work.

**Competing interests:** The authors have declared that no competing interests exist.

## Introduction

Acute physical exercise improves cognitive function [1–3], but physical exercise performed concurrently with a cognitive task – *dual-tasking* – may be even more beneficial [4–7], potentially yielding improvements in both motor [8] and cognitive [9] components.

Executive functions are a set of mental abilities dependent on prefrontal cortex (PFC) activation [10] and comprise three core components: *inhibitory control* (the ability to regulate emotions, thoughts, and behavioural responses), *working memory* (updated temporarily stored information), and *cognitive flexibility* (adapting our behaviour in response to environmental changes; [11]. Collectively, these components may be important for dual-tasking in the real world, for example, when cycling on roads: Inhibitory control is required to suppress prepotent emotional responses to antisocial behaviour, working memory is crucial for recalling locations of other road users prior to manoeuvring, and cognitive flexibility is needed to respond to ever-changing traffic conditions. Moreover, it is possible that these executive functions are developed for these reasons. Hence, it is timely to examine the potential benefits of cycling in combination with cognitive exercise, for executive function, and to explore potential underlying mechanisms for such purported benefits.

Availability of oxygen in the brain is crucial for cognitive function [12,13]. Accordingly, research suggests that improved cognitive task performance following an acute bout of exercise is accompanied by elevated levels of cerebral oxygen saturation ($rSO_2$) in the PFC [12,14,15], a relationship that is partly determined by exercise intensity and duration [15,16]. Recent evidence shows that combined cognitive and cycling exercise increases PFC oxygenation more than cognitive exercise by itself, during both EF and non-EF tasks, albeit not significantly differently from cycling per se [17]. Indeed, several reviews indicate the benefits of cycling exercise in isolation [1,3,18,19], and it is potentially more effective than treadmill running [20] – and safer when considering potential compromise of motor task performance in dual-tasking paradigms [21].

Dkaidek and colleagues [3] conducted a meta-analysis focused on the effects of an acute bout of ergometer cycling on young adults' EF. Their findings showed a positive effect of cycling on EFs, an effect that was moderated by exercise intensity and duration, EF task type and task onset relative to exercise cessation. Dkaidek et al. suggested that moderate cycling exercise intensities (46–64% $VO_{2max}$; [22]) should yield optimal benefits, consistent with the inverted-U hypothesis [23] and Cooper's [24] catecholamine hypothesis – and that this effect appears more pronounced for inhibitory control tasks (e.g., Flanker Task), when the EF task is completed immediately post-exercise. Although there is evidence to suggest that a duration of only ten minutes elevates catecholamine levels, and cognitive performance accordingly [2], Dkaidek and colleagues showed that 20–30 minutes of cycling appear to confer the greatest benefits on EF task performance, consistent with previous reviews [1,2].

When assessing potential EF task enhancements resulting from dual-task exercise, it is important to consider the nature of the cognitive task. Recent evidence suggests that dual-task interventions which incorporate novel and mentally

demanding tasks that engage EFs may be particularly effective [6,17]. Visual foraging tasks (VFTs; [25]) are demanding visual search tasks in which participants must identify multiple dynamic targets from an array of similar moving distracters. There is evidence that performance in VFTs may be associated with better working memory and cognitive flexibility [26]. Multitarget foraging is also representative of visual attention allocation in real-world contexts (e.g., cycling) than traditional single-target visual search tasks, and may consequently provide insights regarding human performance in dynamic environments [27,28], as well as elucidating attentional dynamics and strategies [29,30]. For example, conjunctive foraging – tasks in which more than one target features (e.g., shape + colour) must be identified – necessitate a top-down approach in which eye movements indicate cognitive effort invested in the task [31,32]. Given the PFC's role in top-down control of attention, working memory, and attentional biasing [33,34], it may also be prudent to investigate PFC activation and gaze behaviour during a visual foraging task.

The effects of cycling exercise on subsequent executive function are established [3]. However, the effects of performing cycling and visual foraging in combination are not. Accordingly, the aim of the present study was to investigate the effects of combining ergometer cycling with a visual foraging task (EC + VF) on EF task performance, prefrontal activation and gaze behaviour, when compared to ergometer cycling (EC) or visual foraging (VF) in isolation. In line with recent evidence for the effect of cognitive-motor dual-tasking on cognitive function [4–9], we hypothesized that EC + VF would result in greater EF improvements, and that this would be reflected in increased PFC oxygenation and eye movements. However, given potential dual-tasking interference [21], we also tentatively predicted that VFT task performance may compromise cycle ergometer performance, and that this would be reflected in lower cadences, energetic output, and subjective arousal.

## Method

### Participants and study design

Sample size was estimated using G*Power 3.1 [35]. Estimates were based on a published effect size of $d = 0.52$ for acute moderate intensity cycling on EF task performance [3], experimental power of 0.80, and a significance level of 0.05 for a repeated measures ANOVA (Condition [EC, VF, EC + VF] x Time [pre-, post-intervention), to yield a desired sample size of 17 participants. However, G*Power does not derive estimates for interaction effects for mixed-design repeated measures ANOVAs, so this figure may represent an underestimation. Further, because this was a pilot study – one that required multiple demanding laboratory visits from participants leading to potential participant attrition – a total of 27 participants were recruited. This sample size is also comparable to those used in previous studies examining the effects of dual-tasking [9,17,36–38].

We adopted the 'ANOVA: Repeated measures, within factors' G*Power analysis protocol to approximate the sample size for a within-subjects design. The parameters were as follows: an effect size of Cohen's $f = 0.26$ (converted from Cohen's $d = 0.52$), $\alpha = 0.05$, power = 0.80, one group, and six repeated measurements (derived from 3 conditions x 2 time points). This analysis yielded a required sample size of 17 participants. However, to account for potential underestimation, participant attrition, and comparability with previous studies, we aimed to recruit 27 participants.

During the period 26th May 2023–31st October 2023, twenty-seven healthy young adults (Female = 15; M age = 25.44 ± 4.31 years) were recruited to participate in this study. All participants were volunteers and were recruited via the lead author's institutional intranet, word-of-mouth, posters, and social media platforms – notably, Instagram, Twitter (now X), and WhatsApp Messenger. Participants completed a modified version of the International Physical Activity Questionnaire Short Form (IPAQ short form, 2016; [38,39] during their initial lab visit to assess their physical activity levels. All participants were free of any known cardiovascular, neurological, and pulmonary disorders and had normal, or corrected-to-normal, vision and hearing.

An average metabolic equivalent (MET) score was calculated from the IPAQ data for each category of physical activity. The average value for vigorous physical activity was 1,457.56 ± 2,011.96 MET-minutes per week (Range = 0–6,697.67),

moderate physical activity was 602.61±965.73 MET-minutes per week (Range=0–4,186.05), and walking, 2,241.73±5,166.85 (Range=132–1381.39). The average total physical activity was 4,301.89±5,894.49 MET-minutes per week (Range=132–11,576.72). Accordingly, 11 participants' physical activity level could be classified as *high*, 11 as *moderate*, and 5 as *low*.

Fig 1 illustrates the study design. A repeated measures pre-post design was employed, in which participants provided baseline data during an initial visit to the lab, followed by participation in each of three different experimental conditions, completed on three different occasions.

### Equipment, material and measures

**VO$_{2max}$ testing.** A Lode Excalibur Cycle Ergometer (Cranlea Human Performance Limited, UK) was used to perform VO$_{2max}$ continuous ramp testing, which was preferred to an incremental protocol (see [40]). The participants wore face masks and mouthpieces during testing. During the ramp test, breath-by-breath pulmonary gas exchange data were collected using a Cortex Metalizer 3B gas analyser (Cortex, Leipzig, Germany) and averaged every 30 s.

Participants' height and weight were measured using weighing scales and a stadiometer for data to be inserted into the PC. Heart rate (HR), Blood lactate levels (capillary blood sample), respiratory exchange ratio (RER) and Rate of Perceived Exertion (RPE) data were inputted to Cortex Metalizer 3B analysis software running on a dedicated PC (Dell, Latitude 5290).

**Measures.** Measurements of blood metabolites, cardiovascular parameters and perceived exertion were obtained before and after the VO$_{2max}$ testing (see physiological data in Supplementary Material A in S1 File).

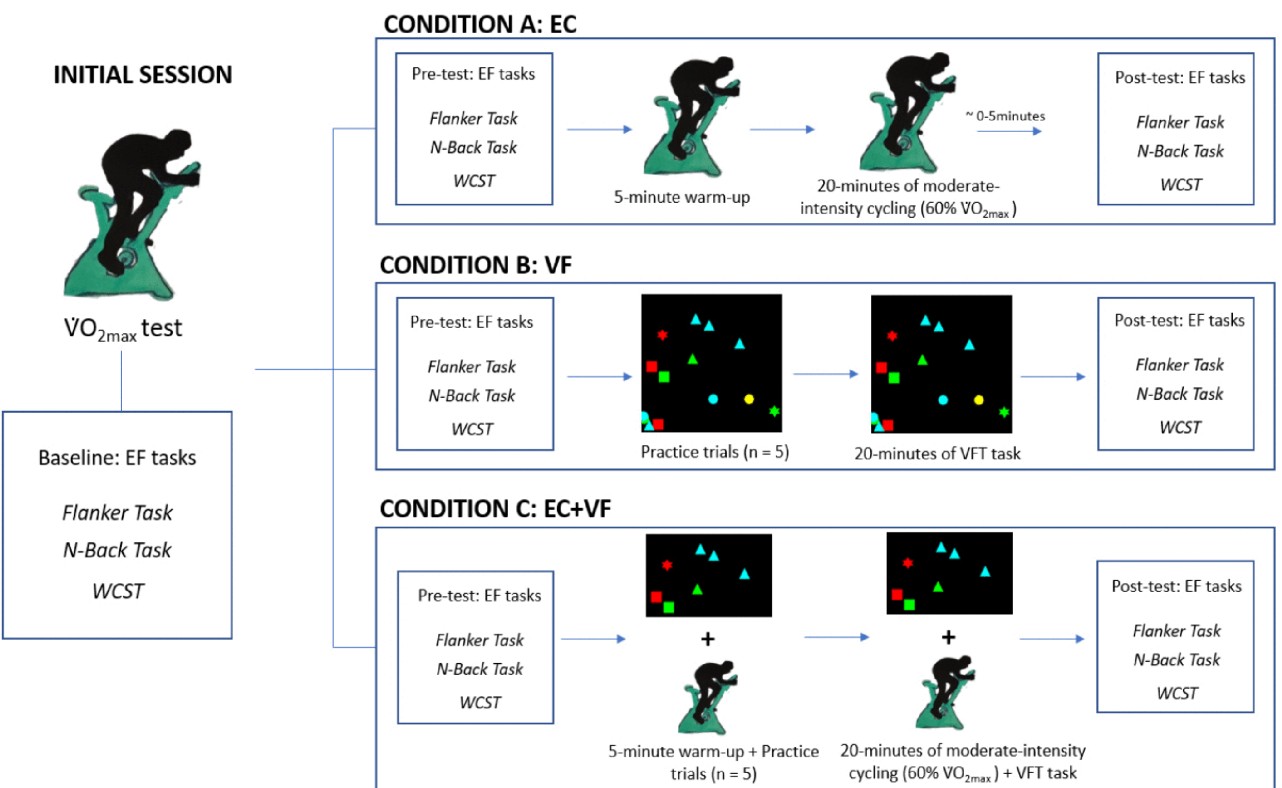

**Fig 1. Study design.**

*Blood metabolites.* Participants' blood lactate levels were taken from the earlobe with a disposable lancet; approximately 0.05 ml of blood was used for analysis using an electronic lactate analyser (Biosen C-Line; EKF Diagnostics Holdings plc, UK) Machine. Capillary Blood samples were taken prior to and immediately post-ramp testing.

*Cardiovascular parameters.* Heart rate measurements were continuously measured via Polar Bluetooth Smart chest strap (Polar Electro Oy, Professorintie, 90440 Kempele, Finland).

*Perceived and actual exertion.* Borg's original Rate of Perceived Exertion scale (Borg's RPE scale; Borg, 1982; https://borgperception.se/) was used to measure the participant's perceived exertion before, and immediately after the exercise. The scale ranges from 6 (no exertion) to 20 (maximal exertion). Also, considering the indeterminate and potential relationships between exercise intensity and cognitive performance [1–3], we assessed *cycling efficiency* – the balance between energy output and input, which is influenced by cycling cadence [41] – via the Lode Ergometry Manager Software (V9).

*EF Tasks.* Participants completed a computerized battery of cognitive EF tasks to assess their inhibitory control (Flanker Task), working memory (2-back version of the n-Back task) and task-switching (Wisconsin Card Sorting Task; WCST) abilities (cf. [42]). Multiple measures of EF were included, as suggested by Etnier and Chang [43]. The tasks were created in Psytoolkit (v. 3.4.2.; [44,45]). The EF tasks were displayed on a monitor (Ilyama ProLite 82280HS) measuring 61.2 cm wide and 47.9 cm high. At a viewing distance of 65 cm, the screen bisected 50.4° of visual angle in the horizontal plane and 40.4° of visual angle in the sagittal plane. Participants provided trial-by-trial responses for each task by pressing keys on a UK QWERTY keyboard.

*Flanker Task.* This task assessed inhibitory control [46]. It comprised 128 trials in which a target letter was presented centrally and flanked on both sides by two distractor letters (cf. [42]; Fig 2). Each set of letters included a central fixation point underneath. Sixty-four of the trials were congruent (i.e., the central letter was identical to the flanking letters) and 64 were incongruent (i.e., they differed). All trials were presented in a randomized order across two blocks of 64 items with a 10-second break between blocks. Participants were instructed to press the 'A' key if the target letter was an X or a C, and to press the 'L' key if it was a V or a B. If the participant pressed the correct key, the central fixation point flashed green for 150 ms and if they pressed the incorrect key, the central fixation points flashed red for 300 ms.

The 'Flanker Effect' was calculated as the difference between the average reaction time of the incongruent and congruent trials. By considering the magnitude of the interference effect, it has been shown to relate to individual difference in PFC engagement, associated with inhibitory control [47,48].

*2-Back Task.* This task assesses working memory [49]. The 2-Back Task comprised 50 trials in which letter stimuli were presented for a duration of 500 ms (cf. [50], bordered by grey horizontal lines at the top and bottom (see Fig 3). Trials were divided into two 25-trial blocks, separated by a 10-second break. For 1 in every 3 trials on average, when cued to do so, the participant was instructed to state whether the current letter was identical to one they saw two trials earlier by pressing the 'M' key; they had 3000 ms to respond before a new stimulus appeared. If the letters were different the participant was instructed not to press any keys. If participants responded correctly, green borders appeared at the top and bottom of the letter (see rightmost image of Fig 3); if they were incorrect, then a red border appeared. Accuracy was calculated as the percentage of hits, otherwise referred to as 'correct matches', using the formula: (hits/(hits + errors)) x 100.

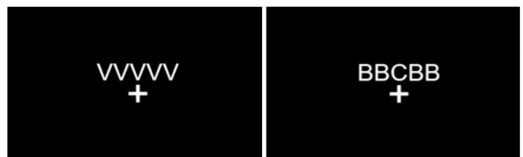

**Fig 2. Flanker task trial examples – congruent (left) and incongruent (right).**

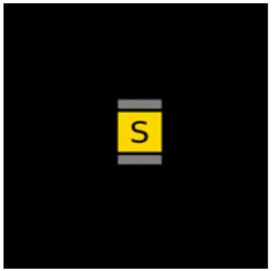 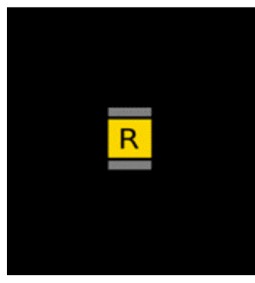 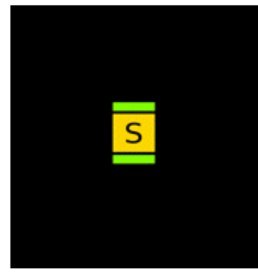

**Fig 3. 2-Back trial example (correct participant response shown in rightmost image).**

Other reported data include the percentage of false alarms, hits that were detected as matches but were not, and missed items, matches that were not identified by pressing the 'M' key.

*Wisconsin Card Sorting Task (WCST).* This task assesses participants' ability to shift attention between different tasks (cognitive flexibility; [51]). This version of the task comprised 64 trials. Four cards were shown at the top of the screen. The participant's aim was to figure out a classification rule that could be used to sort a card displayed in the bottom-left corner of the screen. The participant was instructed to click on one of four cards at the top that they thought belonged to the same category as the bottom-left card; for example, the correct selection in Fig 4 for the rule *'2 items on a card'* would be the second card from the left. After they made their selection, the participant received feedback, but if the rule they adopted was incorrect, they had to make another selection, based on a different rule. Classification rules changed according to the shape of the symbols, the colour of the symbols, or the number of shapes on each card – and the rules changed randomly. Task scores reflect how well participants adapt to these changing rules – i.e., changes in task requirements. The reported raw scores were derived for statistical analysis: Perseverative Errors (i.e., repeated ones) and Non-perseverative Errors (i.e., random ones). Error count was calculated as the sum of all errors, i.e., Non-perseverative Errors + Perseverative Errors.

*Ergometer Cycling.* All cycling was completed on a Lode Excalibur Ergometer (Excalibur Sport, 2006) connected to the Lode Ergometry Manager Software. Participants warmed up for three minutes at 25W. During the intervention

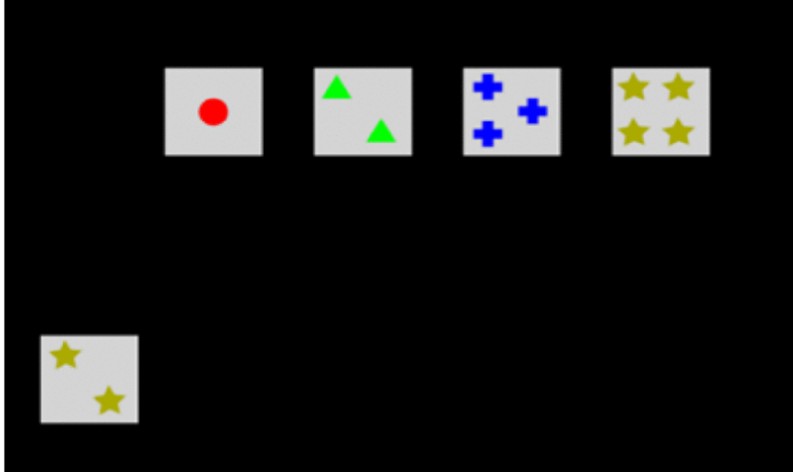

**Fig 4. WCST trial example (rule = 2 shapes OR stars OR gold coloured).**

each participant cycled at the load corresponding to 60% of their $VO_{2max}$. Participants' accumulated energy levels (kJ) and cadence (revolutions per minute; rpm) were continuously recorded throughout, to determine equivalence of effort across the two cycling conditions. Participants were instructed before each condition to maintain a target cadence of 60–70 rpm. Further, exercise intensity was standardized across participants using individualised $VO_{2max}$ testing. As each participant exercised at a fixed percentage of their individual $VO_{2max}$ (60%), we ensured a consistent relative intensity across the conditions. As the resistance remained constant across the exercise conditions and only cadence fluctuated slightly within the instructed range, the relative exercise intensity was controlled. After completing the intervention, participants cooled down for three minutes at warmup intensity. All data were downloaded for analysis in Microsoft Excel.

*Visual Foraging Task (VFT).* The VFT was developed using Clickteam Fusion 2.5, a game development tool that allows for game and software creation of 2D games. Pilot testing was conducted to optimise the VFT for the current purposes: regarding clarity and size of shapes, the display settings, and behaviours. The iterative design process comprised 11 testing phases, during which adjustments were used to augment its overall effectiveness and difficulty level.

The VFT task was designed based on that used by Kristjánsson and colleagues [27]. Our VFT comprised dynamic, multitarget, conjunctive foraging, in an attempt to mimic the complexity of visual search in the real world. Each trial incorporated 40 moving shapes; this number was chosen according to Kristjánsson and colleagues' [28] findings, which implied that forty targets presented a moderate-difficulty challenge relative to set sizes of 20, 60 and 80 shapes; this level was chosen to avoid extensive effort costs; response times increase linearly as set size increases [28].

The VFT was displayed on a projection wall to provide an image that measured 3.90 m in width by 2.17 m in height, 1.68 m away from the participant when seated on the ergometer. It bisected 33.6° of visual angle in the horizontal plane and 49.2° of visual angle in the sagittal plane. An array of 40 moving shapes – specifically, squares, circles, triangles, and stars displayed in different colours (green, red, blue, or yellow) on a black background – were presented to participants, whose aim was to identify specific colour-shape combination target stimuli. Shapes 'rebounded' from the borders of the display in a way that was consistent with the kinematics of physical counterparts. Prior to the start of each trial, the target stimulus would flash on the screen for 3 seconds. If the target was a yellow circle, then the participant would be required to find all instances of yellow circles in that trial (see Fig 5). The minimum number of targets was one, the maximum was six. Each trial lasted 6 seconds and shape-colour target combinations varied from one trial to the next. Participants verbally reported the number of target shapes they think they saw after each trial; responses were scored as correct if their response matched the actual value. Trials were presented in six blocks of 20 trials – a total of 120; there were 25-second breaks between blocks. The percentage of correct trials were calculated for each participant.

*Eye Tracking.* Participants wore Tobii Pro Glasses 2 which were wirelessly connected to a PC running iMotions software (v. 9.3). The glasses were calibrated for each participant using a calibration card held parallel to the eye tracker at a distance of 0.8 to 1.2 meters from the participant, in front of a plain and static background. Participants were instructed to

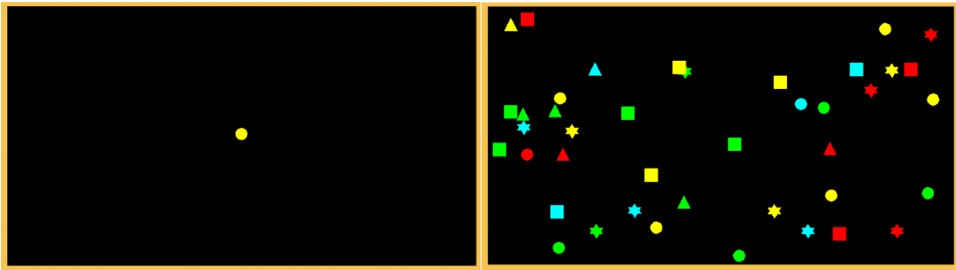

**Fig 5. VFT trial example (yellow circle target; correct answer = 5).**

keep their head still and look at the calibration dot in the middle of the card until calibration was complete. Eye movement data were captured using the iMotions software.

*Functional Near-Infrared Spectroscopy (fNIRS).* Cerebral oxygenation of the PFC was measured using fNIRS (INVOS 5100C Near-infrared Cerebral Oximeter; Somanetics, Troy, MI) via two pads, each housing two optodes, positioned above the eyebrows on the left and right sides of the forehead (i.e., over PFC; cf. [14,52]). Alcohol wipes were used to remove excess sebum from the participant's forehead, which was then dried with a sterile gauze pad prior to taping the pads to the skin to minimise daylight interference. The optodes emitted a signal at 300 and 810 nm wavelengths. The oximeter determined $rSO_2$ by analysing reflected near-infrared (NIR) light at each sensor, to obtain an averaged value every thirty seconds. $rSO_2$ values were not normalised to baseline but were analyzed as absolute $rSO_2$ values. This enabled direct comparison of $rSO_2$ levels across conditions without transformation.

*Affect Grid*. Given the influence of affective state on EF [53,54], we sought to capture participants' affective state using the Affect Grid [55], which is based on Russell's Circumplex Model of Affect [56]. It is a 9-by-9 grid single-item self-report measure of affect that has been used to differentiate affective states in various contexts (e.g., [57–59]). The original circumplex model comprises two perpendicular dimensions – perceived activation (*arousal*) and affective valence (*pleasure-displeasure*) – creating four quadrants that represent *high activation-high valence*, *high activation-low valence*, *low activation-high valence*, and *low activation-low valence* states. Participants indicate their affective state by marking a cross in one square of the Affect Grid.

## Procedure

This study was approved by the Brunel University of London College of Health, Medicine and Life Sciences Research Ethics Committee (Review Ref. 42007-MHR-May/2023− 44830−2). Participants provided their written informed consent to take part in the study. Participants provided their written informed consent and had the opportunity to ask questions before commencing their participation in the study. All participants completed an initial lab visit followed by three conditions in a randomized and counterbalanced order. Consecutive visits were separated by at least 48 hours to minimise the effects of fatigue, and all three conditions were completed within a six-week period (M days between sessions = 8.31 ± 7.29).

Participants were instructed to avoid drinking coffee or alcohol, and strenuous exercise for 24 hours prior to their visit, to abstain from eating in the preceding two hours, and to be adequately hydrated. They were also required not to take any non-essential medication that could affect their physical or cognitive performance in the 12 hours preceding their participation.

*Initial Lab Visit.* Health status was assessed at the initial lab visit via a Health Check Questionnaire and a COVID-19 form. Participants then completed familiarisation trials until they felt confident with the three EF tasks and the VFT, followed by five practice trials. Then, they provided baseline data for the three EF tasks.

*Testing Session.* $VO_{2max}$ data were collected in accordance with the British Association of Sport and Exercise Sciences (BASES; 1997) guidelines, which state that, for measurements to be valid, it must meet three out of the five criteria shown in Table 1. Once the data for the $VO_{2max}$ were collected in the Cortex Metalizer 3B analysis software, the breath-by-breath

**Table 1. BASES $VO_{2max}$ measurement criteria.**

| 1 | A plateau in the $VO_{2max}$ and exercise intensity relationship |
|---|---|
| 2 | A respiratory exchange ratio of 1.15 or above |
| 3 | A final heart rate within 10 beats min per minute of the participant's age predicted maximum heart rate (calculated as 220 – age) |
| 4 | The participant reaches fatigue and volitional exhaustion |
| 5 | An RPE of 19–20 is indicated |

 

data were exported into Microsoft Excel. $VO_{2max}$ was calculated as the average oxygen consumption during the 30-second epoch prior to voluntary termination of the test. Between participants, sterilising solution was used to disinfect the mouth-pieces and face masks to avoid cross-contamination. Each participant's $VO_{2max}$ was used to calculate the appropriate cycle ergometer load for them to perform moderate-intensity exercise – designated as 60% of $VO_{2max}$ (see [3]).

## Experimental conditions

Following the initial lab session, participants completed three experimental conditions: ergometer cycling (EC), visual foraging (VF) or a combination of ergometer cycling and visual foraging (EC + VF); the running order was counterbalanced across participants. At the beginning of each lab visit, the participants completed the Flanker Task, WCST and 2-Back Task; the order in which these were completed was also counterbalanced, across experimental sessions. Eye tracking and fNIRS data were acquired continuously in the VF and EC + VF conditions. fNIRS data were acquired in the EC con-dition, but eye tracking glasses were worn for standardisation purposes and participants were instructed to look as they typically would when performing a stationary activity. In all conditions, participants completed the computerized EF test battery a second time within five minutes of finishing the intervention (cf. [3]).

*Ergometer Cycling (EC).* Participants completed a 3-minute warmup at a 25W load. After, the ergometer load was gradually increased until their oxygen consumption reached 60% of their $VO_{2max}$, at which they continued to cycle steadily for 20 minutes (cf. [1,3]) at a cadence of 60–70 rpm. Participants finished the session with a three-minute cooldown with a resistance of 25W at a self-selected rpm.

*Visual Foraging (VF).* Participants completed five familiarisation trials followed by 20 minutes of VF. Before the end of each trial, participants verbally reported the number of targets they detected, for the researcher to record their response. The participants were informed that each trial was six seconds long and that each trial would commence immediately after its predecessor. If the participants did not report a number for a trial, their response was recorded as incorrect.

*Ergometer Cycling and Visual Foraging (EC + VF).* For the EC condition, participants began with a 3-minute warmup at a 25W load; they simultaneously completed five practice VFT trials. Then, the ergometer load was gradually increased to 60% of their $VO_{2max}$, at which they cycled for 20 minutes at a cadence of 60–70 rpm, while concurrently performing the VFT. Participants finished the session with a cooldown with no resistance and a self-selected rpm. An image of the EC + VF set up is shown in Fig 6.

## Data analysis

All statistical analyses were conducted using IBM SPSS software (Released 2020. IBM SPSS Statistics for Windows, Version 27.0. Armonk, NY: IBM Corp). Where the assumption of sphericity was violated, a Greenhouse-Geisser correction factor was applied. When a significant interaction or main effect was detected, it was followed up by a post hoc pairwise comparison with Bonferroni Correction. For all statistical comparisons, an alpha level of $p < .05$ was used.

*EF Data.* Pre-test and post-test data were expressed as mean scores for the Flanker Task, WCST and 2-Back task. To compare EF task performance in the three conditions, repeated measure analysis of covariance (ANCOVA) was used to determine interactions and main effects in a Condition (EC, VF, EC + VF) x Times (Pre, Post) design with EF baseline scores used as a covariate for the EF outcome measures (cf. [60–62]). Mauchly's Test of Sphericity was used to assess statistical assumption of sphericity and in the case of a violation, Greenhouse-Geisser correction was applied. Bonferroni-corrected pairwise comparisons were run to determine where any group differences lay.

Baseline EF performance was included as a covariate to control for individual differences and improve sensitivity in detecting intervention effects. Although 'Time' captures pre-post changes, adjusting for baseline reduces between-subject variability and potential bias. To address concerns about multicollinearity, we examined correlations between baseline and pre-intervention EF scores across conditions and tasks; all correlations ranged from .10 to .61, below the accepted

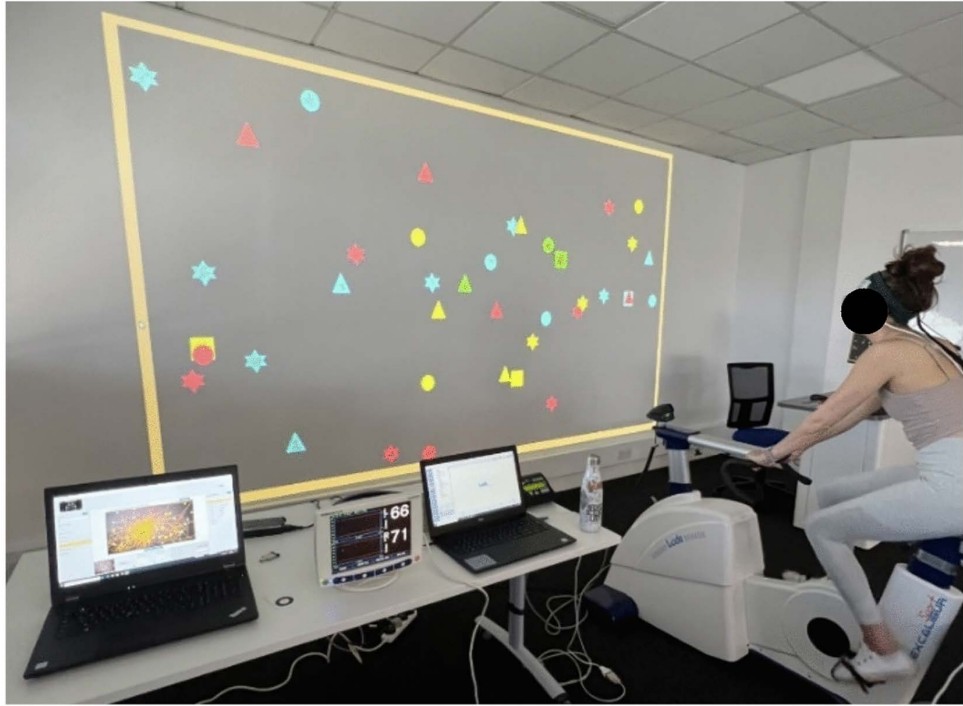

**Fig 6. Experimental setup (EC + VF condition).**

multicollinearity threshold of r = .70 [63]. These results indicate that baseline EF and pre-intervention scores were not redundant, supporting the use of the ANCOVA model.

*Affect Grid Data.* A 3 X 2 (Condition x Time) repeated measures ANOVA was used to examine changes in self-reported affect as assessed using the Affect Grid. *VFT Performance and Gaze Data.* VFT scores were expressed as the mean and were analysed using a paired samples t-test between the EC + VF and VF conditions. The number of gaze fixations was compared across the two conditions, as an index of the extent of visual foraging. Eye movements were recorded using Tobii Pro Glasses 2 at a sampling rate of 100 Hz. The glasses feature a wide-angle scene camera with a resolution of 1920 × 1080 pixels, capturing video at 25 frames per second with a 90-degree horizontal field of view (16:9 aspect ratio; [64]). The glasses were wirelessly connected to a PC running iMotions software (version 9.3). Fixations were automatically detected using iMotions' built-in dispersion-based algorithm. Gaze behavior was analyzed as the number of fixations, which served as an index of visual attention. The mean number of fixations in the two visual foraging conditions was compared using paired samples t-tests.

*fNIRS Data.* $rSO_2$ levels in each condition were expressed as the mean and compared using a repeated measure ANOVA. Correlational analyses were performed to explore potential relationships between (a) VFT scores and $rSO_2$ levels (b) Affect and $rSO_2$ levels and (c) $rSO_2$ levels in left and right PFC.

*Accumulated Energy and Cadence.* Accumulated energy (kJ) in the EC and EC + VF conditions was expressed as the mean and compared using a paired samples t-test. Cadence was calculated as the average rpm across four timepoints throughout the cycling protocol: 5 minutes, 10 minutes, 15 minutes and 20 minutes, then expressed as the mean for each condition, for comparison using a paired samples t-test. Correlations were also performed to examine the relationship between these two measures. Due to missing data caused by hardware malfunction, energy level and cadence data were only available for 21 participants, for both cycling conditions.

## Results

### EF task performance

Table 2 shows descriptive data for all three EF tasks, expressed as means (SDs in brackets) (see also Supplementary Material B in S1 File).

**Flanker task data.** There was no significant condition x time interaction, albeit one approaching statistical significance, $F(2,42) = 2.62$, $\eta_p^2 = .11$, $p = .085$. There was a main effect of condition, $F(2,42) = 3.32$, $\eta_p^2 = 0.14$, $p = 0.046$. Bonferroni-corrected pairwise comparisons revealed no significant differences between pairs of conditions, all $p > .05$. However, there was a significant interaction between condition, time and baseline EF task scores (the covariate), $F(2,42) = 3.54$, $\eta_p^2 = .14$, $p = .038$. To evaluate this interaction, we conducted simple effects analyses using a median split on the baseline scores. Among participants with better performance on the Flanker task at baseline (Median Scores = 17 ms), significant improvement was shown after the EC condition in the Flanker task performance, $t(10) = 3.16$, $p = .01$, d = 0.95, but not the VF or EC + VF conditions ($p > .05$). In contrast, for participants with poorer performance (i.e., lower than the median), no significant change was observed in any of the conditions. These results suggest that EC was particularly beneficial for individuals with higher baseline Flanker Scores.

To explore this interaction further, separate correlations were run for each condition to reveal any relationships between baseline scores and pre-to-post change scores; Fig 7 shows the interaction. These variables were negatively correlated in the EC condition, $r(22) = -.56$, $p = .005$ – as the baseline Flanker Effect increased (i.e., performance worsened), pre-to-post improvements in task performance increased. There were no significant correlations between baseline score and pre-to-post change scores for VF or EC + VF, $p$'s > .05. However, pre-to-post changes in Flanker Task performance for compatible and incompatible trials were positively correlated in all conditions, EC $r(25) = 0.71$, $p < .001$, VF $r(25) = 0.76$, $p < .001$, EC + VF $r(24) = 0.96$, p < .001.

A main effect was found for condition, $F(2,42) = 3.32$, $\eta_p^2 = .14$, $p = .046$; and Mauchly's Test of Sphericity revealed no violations of the sphericity assumption, $W = .96$, $\chi(2) = 1.00$, $p = .607$. However, Bonferroni-corrected pairwise comparisons

**Table 2. EF tasks–Descriptive data.**

| Executive Function Tasks | Baseline | EC | | VF | | EC + VF | |
|---|---|---|---|---|---|---|---|
| | | Pre | Post | Pre | Post | Pre | Post |
| *Flanker Task* | | | | | | | |
| **Compatible RT (ms)** | 670.33 (102.02) | 615.67 (87.83) | 578.74 (99.22) | 622.37 (96.00) | 614.56 (92.51) | 632.72 (99.81) | 606.23 (99.15) |
| **Incompatible RT (ms)** | 693.04 (98.06) | 655.19 (94.59) | 602.15 (95.30) | 650.85 (85.85) | 634.04 (93.03) | 645.33 (85.17) | 625.77 (93.48) |
| **Flanker Effect (ms)** | 22.25 (51.64) | 39.44 (28.52) | 23.41 (23.55) | 28.48 (36.55) | 19.48 (31.58) | 18.31 (32.92) | 19.54 (21.67) |
| *WCST* | | | | | | | |
| **Error Count** | 19.00 (6.54) | 13.81 (6.34) | 14.93 (5.79) | 16.78 (15.90) | 15.26 (5.28) | 15.52 (5.14) | 14.36 (4.71) |
| **Perseveration error count** | 11.00 (3.56) | 8.81 (3.18) | 9.48 (3.37) | 8.11 (6.14) | 8.85 (2.66) | 9.72 (2.82) | 9.52 (3.12) |
| **Non- Perseveration error count** | 7.96 (4.38) | 5.46 (3.48) | 5.41 (3.35) | 8.19 (14.99) | 6.22 (4.38) | 5.76 (3.57) | 5.08 (2.86) |
| *2-Back Task* | | | | | | | |
| **Total trials with a match** | 18.80 (4.86) | 16.22 (3.86) | 15.42 (3.28) | 15.04 (2.79) | 15.07 (3.05) | 15.21 (4.36) | 15.19 (3.16) |
| **Total trials without a match** | 41.96 (10.86) | 33.41 (4.47) | 33.50 (5.95) | 34.96 (2.79) | 34.93 (3.05) | 35.35 (6.42) | 33.92 (5.04) |
| **Number of correct matches** | 13.32 (5.94) | 14.70 (4.30) | 14.31 (3.73) | 13.41 (3.94) | 13.44 (4.10) | 14.17 (4.46) | 14.23 (3.81) |
| **Number of missed items** | 5.27 (5.87) | 1.48 (1.70) | 0.81 (1.17) | 1.63 (2.68) | 1.33 (2.30) | 1.48 (2.43) | 0.96 (1.43) |
| **Number of false alarms** | 1.92 (2.45) | 2.44 (3.42) | 4.35 (13.28) | 1.96 (2.91) | 2.41 (4.24) | 2.36 (2.64) | 2.19 (2.02) |
| **Percentage of correct matches** | 73.80 (25.97) | 89.00 (13.31) | 92.75 (14.76) | 88.87 (19.33) | 91.33 (14.47) | 88.88 (17.29) | 93.02 (11.47) |
| **Percentage of missed items** | 26.24 (25.95) | 11.13 (13.28) | 4.90 (7.97) | 11.39 (19.40) | 8.80 (14.46) | 11.48 (17.32) | 6.90 (11.68) |
| **Percentage of false alarms** | 5.61 (7.05) | 7.43 (10.30) | 4.96 (4.73) | 5.72 (10.06) | 6.83 (12.36) | 6.88 (7.03) | 5.88 (5.81) |

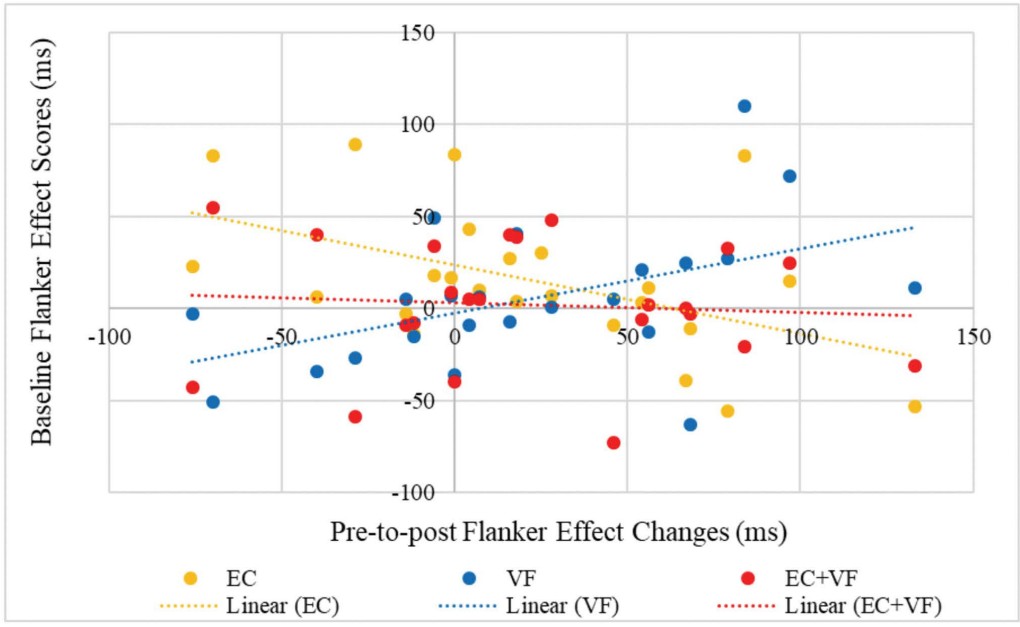

**Fig 7. Flanker effect condition * time * covariate interaction.**

revealed no significant differences between EC and VF, $t(21) = 1.36$, $SE = 6.26$, $p = 0.563$, EC and EC+VF, $t(21) = 2.18$, $SE = 5.71$, $p = 0.123$ nor VF and EC+VF, $t(21) = 0.57$, $SE = 6.92$, $p = 1.00$.

**WCST data.** *Error Count.* A repeated measures ANCOVA did not reveal a condition x time interaction for error count, $F(2,28) = 1.09$, $\eta_p^2 = .07$, $p = .35$, nor main effects of condition, $F(2,28) = .01$, $\eta_p^2 = .00$, $p = .991$, or time $F(2,28) = .08$, $\eta_p^2 = .01$, $p = .784$.

**2-Back task data.** *Correct Matches.* A repeated measures ANCOVA revealed no significant condition x time interaction, $F(2,22) = 1.78$, $\eta_p^2 = .08$, $p = .181$. However, there was a main effect of time, $F(2,22) = 8.06$, $\eta p^2 = .27$, $p = .010$; Bonferroni-corrected pairwise comparison revealed that the number of correct matches increased from pre- to post-intervention, $t(22) = 3.08$, $SE = 1.18$, $p = 0.005$.

## PFC oxygenation

*Left PFC.* The upper panel of Fig 8 illustrates average $rSO_2$ at the left optode, by Condition. Average baseliner$SO_2$ was 69%±2% across all conditions. Average $rSO_2$ was 68%±2% during EC+VF, 66%±0.6% during VF, and 72%±3% during EC.

Mauchly's Test revealed a violation of the sphericity assumption, $W = .27$, $\chi(2) = 49.32$, $p < .001$; hence, a Greenhouse-Geisser adjustment was applied, $\varepsilon = .58$. A main effect was found for condition, $F(1.16,45.17) = 94.18$, $\eta_p^2 = .71$, $p < .001$. Bonferroni-corrected pairwise comparison indicated that EC resulted in greater left PFC activation than EC+VF, $t(39) = 22.23$, $SE = .18$, $p < .001$, and VF, $t(39) = 9.21$, $SE = .47$, $p < .001$. However, *t*here was no significant difference be*t*ween EC+VF and VF, $t(39) = .60$ $SE = .35$, $p = 1.00$.

*Right PFC.* The lower panel of Fig 8 illustrates average $rSO_2$ at the right optode, by Condition. Average baseline $rSO_2$ was 64%±8% across all conditions. Average $rSO_2$ was 66%±2% during EC+VF, 66%±1% during VF, and 70%±3% during EC.

Mauchly's Test revealed a violation of the sphericity assumption, $W = .43$, $\chi(2) = 32.47$, $p < .001$; hence, a Greenhouse-Geisser adjustment was applied, $\varepsilon = .64$. A main effect of condition was found, $F(1.27,162.19) = 156.70$, $\eta_p^2 = .80$, $p < .001$.

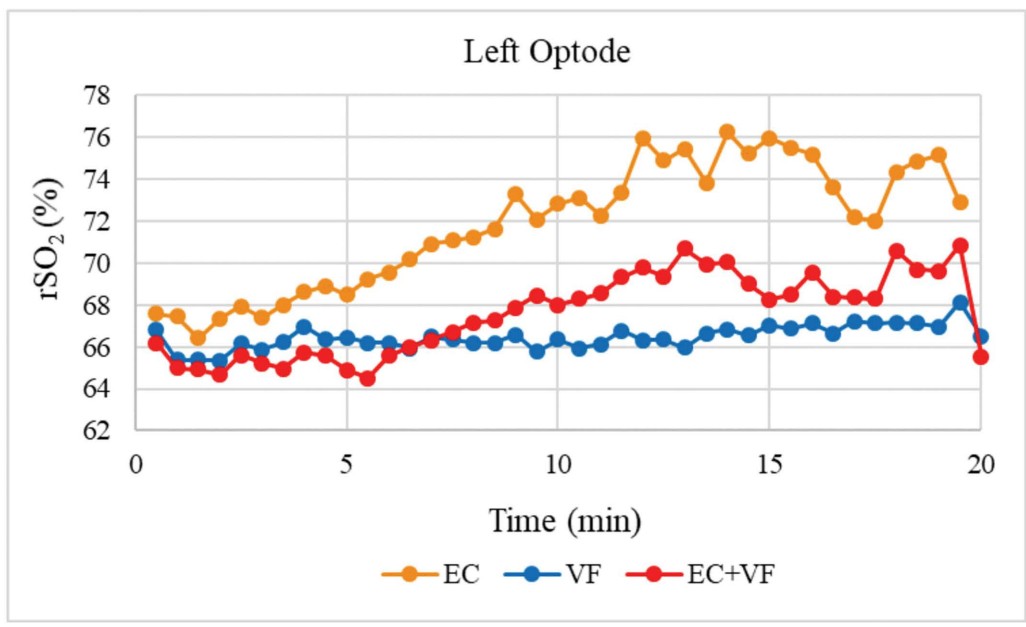

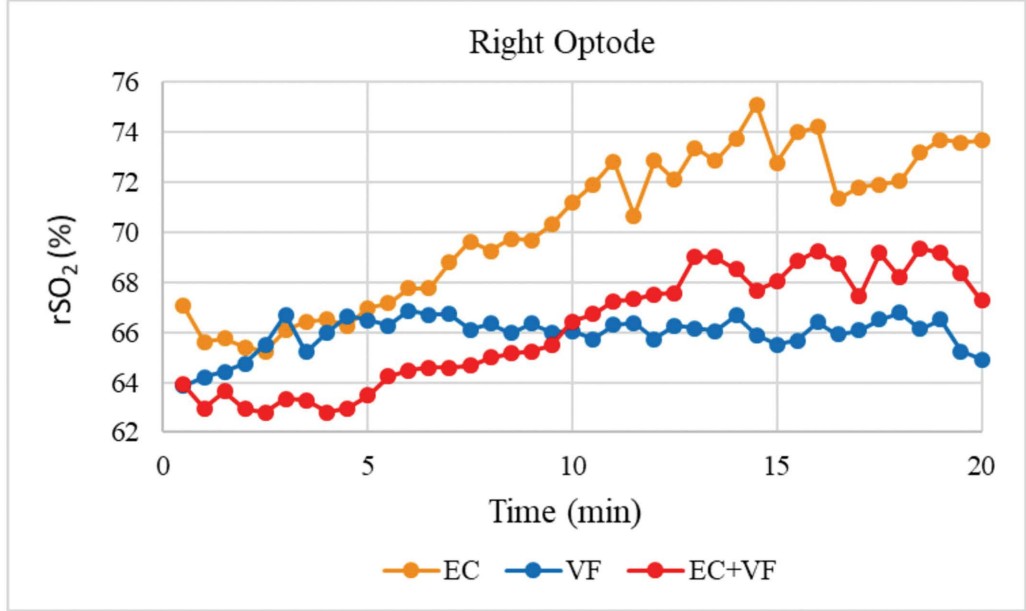

**Fig 8. rSO$_2$ by condition.**

Bonferroni-corrected pairwise comparisons indicated that EC resulted in significantly greater right PFC activation than EC+VF, $t(39)$ = 18.00, $SE$ = .24, $p$ < .001, and VF, $t(39)$ = 12.66, $SE$ = .43, $p$ < .001. Further, right PFC activation was significantly higher in EC+VF compared to VF, $t(39)$ = 4.13, $SE$ = .27, $p$ < .001.

*Left and Right PFC rSO$_2$ Levels – Correlations.* There were positive correlations between left and right PFC rSO$_2$ Levels for EC, $r(40)$ = .94, $p$ < .001, and EC+VF, $r(40)$ = .89, $p$ < .001. However, there was no correlation revealed between left and right PFC for VF, $r(40)$ = .23, $p$ = .159.

An exploratory analysis was conducted to examine potential relationships between rSO$_2$ and behavioral outcomes, but none emerged (all $p$'s > .05).

### VFT Task – performance, PFC activation and gaze behaviour

Fig 9 illustrates relationships between average rSO$_2$ levels and the number of gaze fixations, for both visual foraging conditions. There was no main effect of condition on VFT task performance and no significant difference between the number of gaze fixations in EC+VF and VF, $t(21)$ =.97, $SE$ = 202.90, $p$ = .335 (see Supplementary Material C; see also Supplementary Material D in S1 File). However, there were significant correlations between the number of fixations and right rSO$_2$ levels, $r(23)$ =.46, p = .029, and left rSO$_2$ levels, $r(23)$ =.51, $p$ = .014, during EC+VF. There was no such

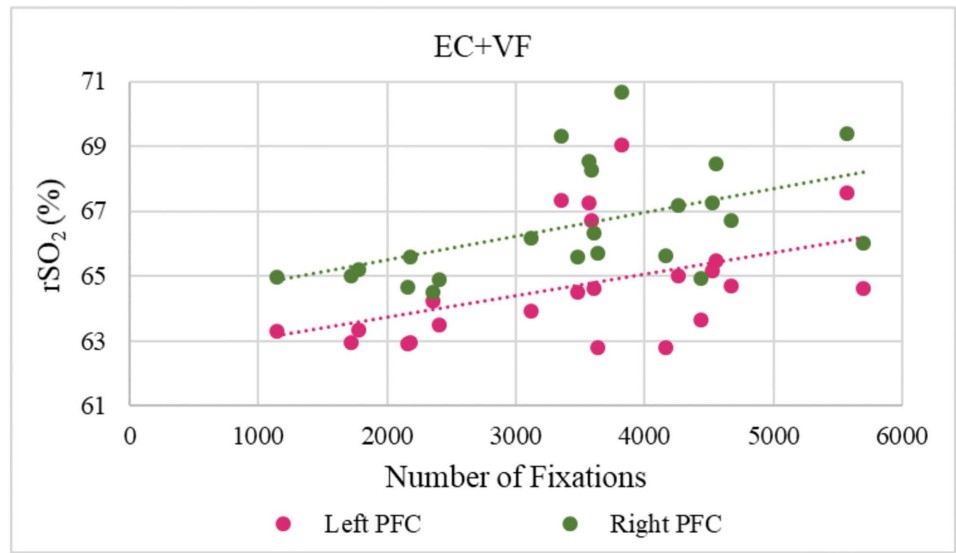

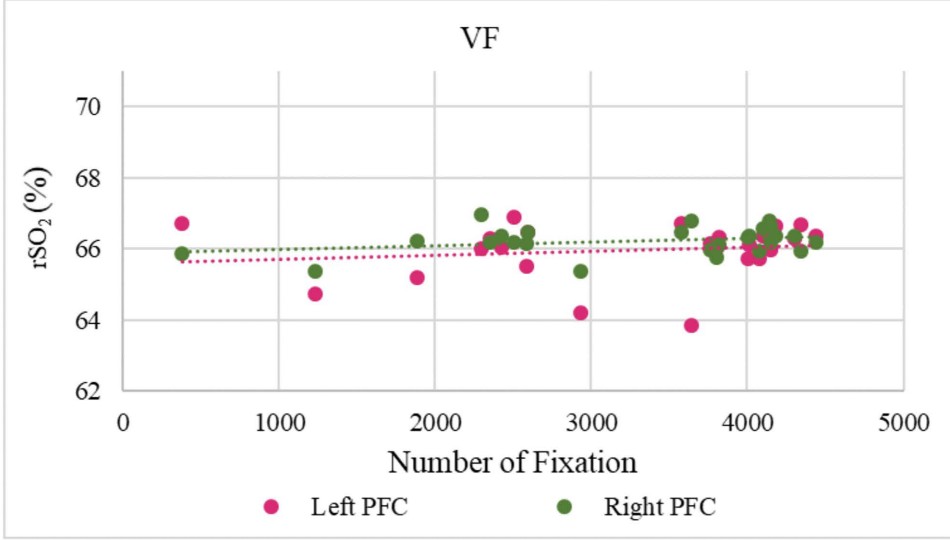

**Fig 9. Correlations between rSO$_2$ levels in the left and right PFC and number of gaze fixations, by Condition.**

relationship between the number of fixations and right rSO$_2$ levels, $r(24) = .28$, p = .166, or left rSO$_2$ levels, $r(24) = .16$, p = .425, during VF.

**Affect grid data**

Affect Grid data are shown in Fig 10.

*Arousal.* There was a significant condition x time interaction, $F(2,50) = 4.89$, $\eta_p^2 = .16$, p = .012; Mauchly's Test of Sphericity did not reveal any violations of the sphericity assumption, $W = .95$, $\chi(2) = 1.73$, p = .554. Bonferroni-corrected pairwise comparisons revealed that self-reported arousal levels increased significantly after EC + VF compared to VF, $t(25) = 4.25$, $SE = .42$, $p < .001$. A main effect of condition was also revealed, $F(2,50) = 4.93$, $\eta_p^2 = .17$, p = .011. Bonferroni-corrected pairwise comparisons revealed that self-reported arousal was greater at both time points, on average in the EC condition, compared to the VF condition, $t(25) = 2.88$, $SE = .37$, p = .024.

*Pleasantness.* There was no significant condition x time interaction, although it approached significance, $F(2,50) = 2.49$, $\eta_p^2 = .09$, p = .093. There was no main effect of condition, $F(2,50) = .06$, $\eta_p^2 = .00$, p = .944. However, a main effect was found

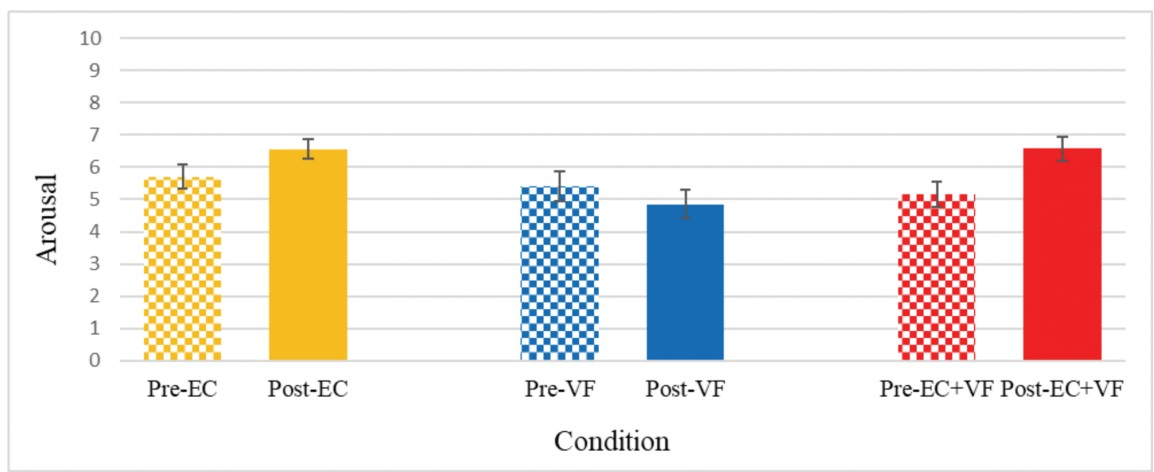

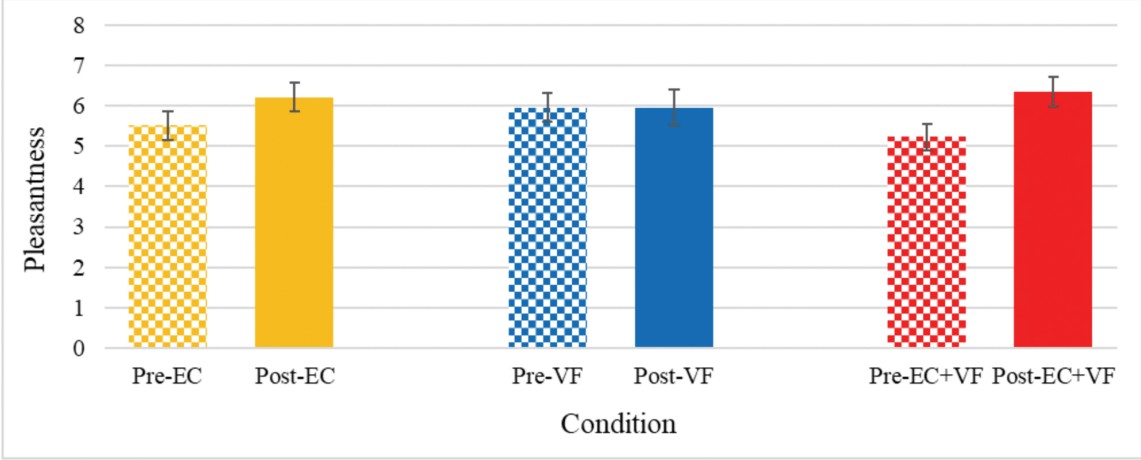

**Fig 10. Mean (± SE) self-reported affect over time, by Condition.**

for time, $F(1,25) = 6.07$, $\eta_p^2 = .20$, $p = .021$. Bonferroni-corrected pairwise comparisons revealed that self-reported pleasantness increased over time, $t(25) = 2.46$, $SE = .26$, $p = .021$.

*Affect and rSO$_2$ levels.* There were no significant correlations between affect and PFC rSO$_2$ levels, all $p$'s > .05.

**Accumulated energy and cadence levels by condition.** Fig 11 illustrates average energy levels and cadence in the EC and EC+VF conditions. A paired samples t-test revealed significantly greater accumulated energy during EC than during EC+VF, $t(16) = 2.31$, $p = .035$. A separate paired samples t-test also revealed that average cadence was higher during EC than in EC+VF, $t(14) = 2.78$, $p = .015$. Correlations revealed a moderate correlation between accumulated energy and cadence during both cycling conditions, $r(17) = 0.60$, $p = .007$. Considering this relationship, and the differences between conditions, plus the differential effect of the cycling conditions on PFC oxygenation, we performed exploratory correlations to investigate potential relationships between accumulated energy and cadence with rSO$_2$ levels. There were no significant correlations between accumulated energy and rSO2 in the EC or EC+VF conditions, $p$'s > .05. In the EC condition, cadence and rSO$_2$ levels in left PFC, $r(17) = .73$, $p < .001$, and right PFC, $r(17) = .69$, $p = .001$, were moderately correlated. There were no significant correlations between cadence and rSO$_2$ in the EC+VF condition, $p$'s > .05.

## Discussion

The aim of this pilot study was to determine whether dual-task exercise comprising ergometer cycling and a visual foraging task (EC+VF) might enhance EF task performance, relative to ergometer cycling (EC) or visual foraging (VF) in isolation. Ergometer cycling, in both conditions (EC and EC+VF), increased PFC oxygenation and subjective arousal

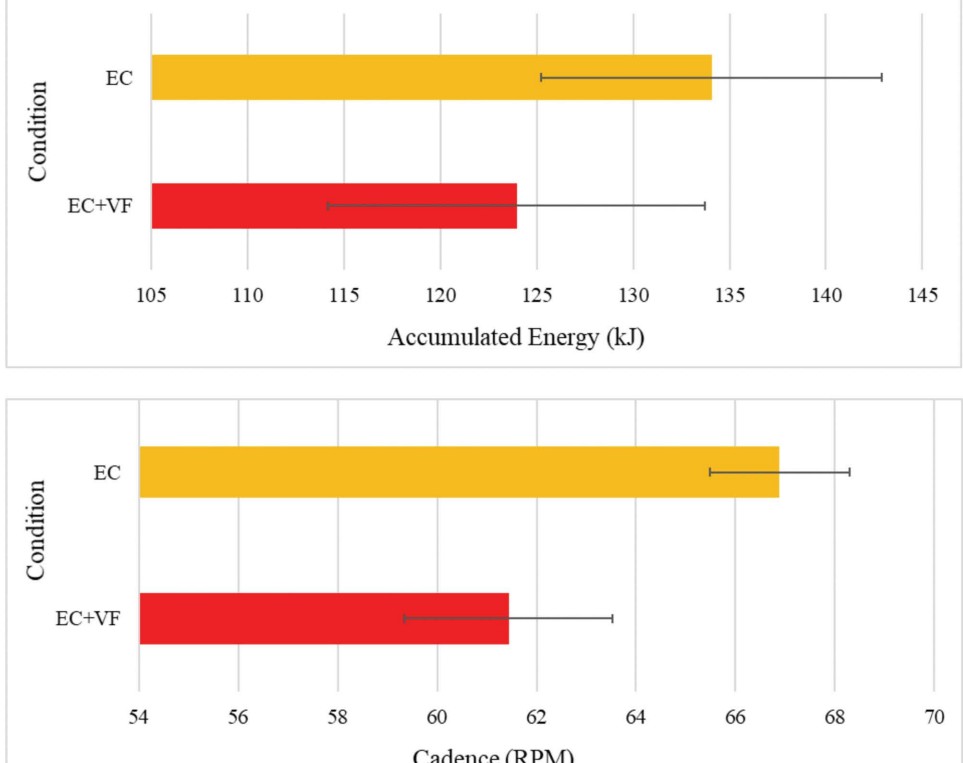

**Fig 11. Mean (± SE) accumulated energy (kJ) and cadence (rpm), by condition.**

more than VF alone, particularly so in the EC condition, and participants with poorer baseline inhibitory control benefited most. Moreover, in the EC + VF condition, rSO$_2$ levels in the left and right PFC were correlated with the number of gaze fixations. However, these changes were not accompanied by improvements in EF task performance. Nonetheless, the condition x time x covariate interaction for the Flanker task suggests that participants with inferior baseline Flanker task performance may benefit the most from the dual-task intervention. Relatedly, Ishihara and colleagues' [65] review showed that, for children who performed well in cognitive tasks at baseline, the benefits of acute exercise interventions were not as pronounced. Based on this association our findings may suggest that, in future work, researchers should consider individual differences in EF.

Our findings suggest that the EC and EC + VF conditions elicited higher subjective arousal than the VF condition, but contrary to our predictions, there was no reduction in arousal in the combined condition. However, the increased arousal levels in this present investigation were not associated with improvements in EF task performance, contrary to previous suggestions [2]. Relatedly, Hacker and colleagues [66] examined the relationship between ergometer cycling durations, EF task performance and subjective arousal and found no relationship between the latter and information processing speed; they suggested that heightened arousal levels might not promote neural adaptations required for cognitive performance improvements. The authors also suggested that increases in cerebral blood flow could be a potential mechanism for exercise-induced cognitive improvements; contrarily, subcortical mechanisms appear to determine subjective arousal (e.g., the amygdala; [67]). Relatedly, there were no correlations between pre-to-post changes in self-reported arousal and PFC rSO$_2$ in the present study.

PFC rSO$_2$ levels increased significantly in the EC and EC + VF conditions relative to the VF condition, but this did not translate into EF task performance improvements (cf. [68]). Exercise-induced improvements in EF performance have been associated with increased cerebral blood flow, which appears to improve frontoparietal EF network efficiency [69] – which can be improved with long-term training. For example, Liu and colleagues [70] reported that 12 weeks of moderate intensity cycling improved participants' Trail Making Task performance, including young adults at their cognitive peak. Byun and colleagues reported similar findings, showing that cortical activation in the left dorsolateral PFC (lDLPFC) increased after cycle ergometer exercise, which also corresponded with improved cognitive task performance.

Our findings also showed that PFC rSO$_2$ increased throughout the course of the cycling intervention (EC and EC + VF), although the increases over time were more linear in left PFC; changes in right PFC were more variable, as reflected in the absence of a correlation between left and right PFC rSO$_2$ values. Left DLPFC is involved in performance of working memory-based executive tasks [71]. However, unlike during the VF condition, both cycling conditions elicited bilateral prefrontal activation, and left and right PFC rSO$_2$ levels were correlated accordingly. Left DLPFC is strongly implicated in top-down control of attention [72], and arises from acute exercise bouts [68,73]; hence, we might have expected to see such lateralisation during the combined condition, but this was not the case.

The increases in PFC activation during the EC condition may reflect greater cerebral blood flow resulting from higher exercise intensities [74,75]. Despite the overt requirement for participants to maintain a cadence of 60 rpm in both cycling conditions, average cadences and accumulated energy were higher in the EC condition than in EC + VF (NB: the two are closely related; [76]. It is possible that, in the combined condition – a dual-task paradigm, effectively – the VF task acted as a distractor from the physical task, thereby impeding performance of the latter; such trade-offs are commonly observed in studies of dual-task performance [77,78]. Relatedly, we observed correlations between cadence and bilateral rSO$_2$ levels in the EC condition, lending support to the notion that increased blood flow when cycling might have facilitated oxygen turnover. Furthermore, the EC condition elicited greater increases in PFC rSO$_2$, cadence and accumulated energy than the EC + VF condition, suggesting higher exercise intensities (cf. [79]), albeit ones that were not reflected in participants' subjective arousal. The above finding could also partly explain the observed correlation between PFC rSO$_2$ levels and the number of gaze fixations in the EC + VF condition. Increased oxygen demand in PFC due to greater top-down control of visual attention arguably elicits smaller changes in blood flow than the more intense exercise stimulus, which

causes macroscopic vasodilation in both brain and body [80,81]. Hence, when we considered the reduced cadence and accumulated energy in the EC + VF condition, there is greater scope for the VFT task to exert effects on prefrontal oxygen demand.

Our complex VF task, which comprised 40 similar and sometimes overlapping shapes, possibly increased interference between target stimuli and distractors – a phenomenon that increases frontal lobe-mediated top-down control [82]. This aligns with the attentional allocation required during on-road cycling, where individuals must continuously monitor and prioritize relevant stimuli amid distractions (e.g., traffic, pedestrians). Thus, the ecological validity of the foraging task is supported by the similarity of these attentional demands. However, to more fully establish its validity, future studies should employ foraging tasks that more closely replicate complex environments and the perceptual load of road conditions. Further, the reductions in cadence and energy output may reflect dual-task costs, consistent with Kahneman's Capacity Model of Attention [83], which suggests that cognitive and physical tasks compete for limited attentional resources. Therefore, the demands of the EC + VF task may have exceeded available attentional capacity, leading to lower cadence levels as participants may have prioritized the VF over EC. Additionally, the correlation between PFC activation and gaze fixations in the EC + VF condition may indicate demands on a shared neural resource or compensatory efforts to maintain performance. More temporally or spatially precise neurophysiological data, such as those obtained via EEG and fMRI data, respectively, would help to elucidate such mechanisms.

Another limitation of this study may be the selected EF tasks. The complexity of EF tasks may influence the quality of the associated outcome measures [84], hence, employing more sophisticated tasks may yield larger effects [84–86]. Our findings suggest that the EF tasks were relatively low in complexity, resulting in a near-ceiling effect in participants' performance. For example, average pre-test scores for the 2-Back and Flanker tasks were 89% and 96%, respectively, leaving limited scope for improvement. Hence, higher-order EF task measures, such as those requiring reasoning and problem-solving [66], may be more appropriate. Utilizing more challenging tasks may minimize ceiling effects, enhance sensitivity, and engage multiple executive function domains, allowing for the detection of subtle changes that simpler tasks may not detect [87,88]. Further, Dkaidek et al.'s [3] systematic review and meta-analysis found the greatest benefits for response time in EF tasks, such as the Flanker task, following moderate-intensity cycling for a duration of 21–30 minutes. Although our intensity was within the moderate intensity range (46–63% VO2max), the 20-minute duration fell marginally below their suggested optimal range which may have attenuated potential effects. Also, the meta-analysis examined intensity and duration as independent moderators, rather than an interaction, leaving open the possibility that their combined influence may differ from their individual effects. Without assessing this interaction, we can only infer that the exercise parameters in the present study did not align optimally, which may have partly contributed to null effects observed.

Our rationale for collecting fNIRS data during all three conditions was to examine how cortical oxygenation varies according to physical and cognitive demands, in isolation and combined. Therefore, our analysis focused on within-subject relative changes in rSO$_2$ to account for individual variability. A limitation is that the fNIRS device is susceptible to extracranial contamination, such as skull thickness and posture with no clear algorithm to correct measurements [89]. However, we took steps to mitigate these confounding factors. These included maintaining consistent participant positioning by placing the fNIRS device after the participant was seated on the ergometer, instructing participants to minimize head movement and to perform the visual search task using eye movements rather than head movements. Additionally, we attempted to reduce external noise by turning off the room lights and closing the window blinds.

## Conclusion

Individual differences may mediate exercise-induced improvements in inhibitory control following a brief bout of moderate intensity cycling. However, the attentional demands of visual foraging while cycling may reduce attention to the motor task and reduce participants' energetic output accordingly. Consistent with previous studies, ergometer cycling increased

self-reported arousal and prefrontal oxygenation, although these two metrics were uncorrelated. Given the relationship of rSO$_2$ with gaze behaviour in the combined condition only, we tentatively propose that whilst physical exercise increases the supply of oxygen to PFC, increases top-down attentional demands during visual foraging tasks increase the demand for that oxygen.

## Supporting information

**S1 File. This Excel spreadsheet contains raw data relating to participants' VO2max test outputs, EF task performance, Affect Grid ratings, VFT task scores, bilateral prefrontal blood flow (fNIRS), gaze fixations, and ergometer cadence and energetic output.**
(XLSX)

## Author contributions

**Conceptualization:** Tamara S. Dkaidek, David P. Broadbent, Daniel Tony Bishop.

**Data curation:** Tamara S. Dkaidek.

**Formal analysis:** Tamara S. Dkaidek, David P. Broadbent, Andre J. Szameitat, Daniel Tony Bishop.

**Investigation:** Amelia Dingley, Tamara S. Dkaidek, Daniel Tony Bishop.

**Methodology:** Amelia Dingley, Tamara S. Dkaidek, Justin Parsler, David P. Broadbent, Andre J. Szameitat, Daniel Tony Bishop.

**Project administration:** Amelia Dingley, Tamara S. Dkaidek, Daniel Tony Bishop.

**Resources:** Amelia Dingley, Tamara S. Dkaidek, Justin Parsler, Andre J. Szameitat.

**Software:** Justin Parsler.

**Supervision:** Amelia Dingley, David P. Broadbent, Andre J. Szameitat, Daniel Tony Bishop.

**Validation:** Tamara S. Dkaidek.

**Visualization:** Tamara S. Dkaidek, David P. Broadbent, Daniel Tony Bishop.

**Writing – original draft:** Tamara S. Dkaidek, David P. Broadbent, Andre J. Szameitat, Daniel Tony Bishop.

**Writing – review & editing:** Amelia Dingley, Tamara S. Dkaidek, Justin Parsler, David P. Broadbent, Andre J. Szameitat, Daniel Tony Bishop.

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
