## [Decision Letter · Decision Letter 0]

19 Jun 2025

Dear Dr. Bishop,

We look forward to receiving your revised manuscript.

Kind regards,

Takao Yamasaki

Academic Editor

PLOS ONE

Journal Requirements:

Reviewers' comments:

Reviewer's Responses to Questions

**Comments to the Author**

1. Is the manuscript technically sound, and do the data support the conclusions?

Reviewer #1: No

Reviewer #2: Partly

2. Has the statistical analysis been performed appropriately and rigorously?

Reviewer #1: No

Reviewer #2: Yes

3. Have the authors made all data underlying the findings in their manuscript fully available?

Reviewer #1: Yes

Reviewer #2: Yes

4. Is the manuscript presented in an intelligible fashion and written in standard English?

Reviewer #1: Yes

Reviewer #2: Yes

Reviewer #1: General Comments

The present study examined the impact of ergometer cycling with visual foraging (EC + VF) on executive function (EF) and prefrontal cortex activation, in comparison with ergometer cycling (EC) and visual foraging (VF) alone. The research hypothesis was that the EC + VF condition would exhibit a more pronounced effect on EF and prefrontal cortex activation than either EC or VF alone. The research theme is interesting, and the findings may contribute to our understanding of the relationship between acute exercise and cognitive function. However, there are several serious concerns regarding the methods that the authors should address.

1) Sample Size Calculation via Power Analysis

The authors state that a power analysis using G*Power indicated an adequate sample size (Lines 122–127). However, the statistical model used in the study—a two-way repeated measures ANCOVA—is not well-supported by G*Power. In general, G*Power does not allow for accurate power analysis for repeated measures ANOVAs involving multiple within-subject factors, as these models include multiple random effects that are not easily distinguishable from error. In fact, G*Power version 3.1 does not include an option for computing within-within interactions for F-tests. The authors should provide a more detailed explanation of how the sample size was determined for the two-way repeated measures ANCOVA.

2) Treatment of Confounders in fNIRS Data

In Lines 284–291, the features of the fNIRS device (INVOS 5100C) are described. However, the authors do not mention how potential confounders in fNIRS data, such as systemic physiological factors (e.g., respiration rate, heart rate), changes in scalp blood flow, and postural changes, were handled, even though it is well known that fNIRS data are vulnerable to these confounders. More details about fNIRS data processing should be provided. Additionally, it is unclear whether cerebral oxygenation was measured during the cognitive tasks. Typically, when investigating the association between cognitive functions and brain activation, fNIRS records oxyHb and deoxyHb during the cognitive tasks, not during exercise itself. Referring to Figure 10, the fNIRS data, represented as oxygen saturation (%), appear to have been measured during both exercise and VF conditions. The authors should clarify the validity and interpretation of fNIRS measurements in this study.

3) Statistical Analyses

The authors used a two-way repeated measures ANCOVA with baseline EF performance as a covariate. However, this model may not be appropriate. Since both “Condition” (EC, VF, EC + VF) and “Time” (Pre, Post) are within-subject factors, the interaction between these two factors is sufficient to test for differential effects on EF. Including baseline EF performance as a covariate may be redundant. Moreover, because baseline scores are likely to be correlated with both pre- and post-test scores, their inclusion as covariates could introduce multicollinearity and bias the results. A repeated measures two-way ANOVA would likely be more appropriate.

4) Explanation of Gaze Fixation

Gaze fixation is a central variable in the current study. However, the description is insufficient; the manuscript only states that “Gaze data were automatically collected into the iMotions software” (Line 380). To ensure reproducibility using different devices or software, the authors should provide more detailed information about how gaze fixation was measured and defined.

5) Other Issues (Typos and Errors)

There are numerous typographical and formatting errors throughout the manuscript. A thorough review and revision are recommended. Examples include:

-Line 154: “(see (41)” -> should be either “(see [41])” or “(see (41))”

-Line 216: “(hits/hits + errors) x 100” → should be “(hits / (hits + errors)) × 100”

-Line 341: “(see 3)” -> it is unclear what this refers to.

-Line 403: “F(2,42) = ηp2 = 0.14, p = 0.046” → the F-value appears to be missing.

-Line 437: “The upper panel of Figure 8 illustrates average rSO2 at the left optode, by Condition.” -> Figure 8 appears to display self-reported affect values, not rSO2.

-Line 445: “t(39) = 22.23.” -> Please verify the degrees of freedom; this value seems unusually high.

Reviewer #2: This pilot study examines the effects of dual-task training (ergometer cycling + visual foraging) on executive function (EF), prefrontal cortex (PFC) activation, and gaze behavior. The topic is timely, and the experimental design is rigorous, incorporating fNIRS, eye-tracking, and validated EF tasks. While the findings contribute to the literature on exercise-cognition interactions, several issues require clarification to strengthen the manuscript.

Major Comments

1. The authors hypothesized that the combined condition (EC+VF) would yield greater EF improvements, yet the results did not support this (e.g., near-ceiling effects in EF tasks). This discrepancy should be discussed in more depth, addressing potential explanations such as task difficulty or individual differences. Consider higher-complexity EF tasks (e.g., 3-Back, Stroop) in future work to avoid such constraints.

2. The reduced cadence/energy in the EC+VF condition suggests dual-task costs, but the mechanistic explanation (e.g., resource competition vs. strategic trade-offs) is underdeveloped. The authors should: Cite dual-task frameworks (e.g., Kahneman’s Capacity Model) to contextualize findings. Discuss whether the observed PFC-gaze correlation in EC+VF reflects shared resources or compensatory effort.

3. Exercise Intensity Control: Cadence varied significantly between EC and EC+VF (Fig. 11). Was intensity matched via HR or RPE? If not, this confounds comparisons. fNIRS Data: Specify whether rSO₂ changes were normalized to baseline. Correlations between rSO₂ and behavioral outcomes (beyond gaze) would bolster claims about neural efficiency.

4. The Condition × Time × Covariate interaction for the Flanker task (p = 0.038) is a key finding but lacks post-hoc detail. Include effect sizes (e.g., η²) and simple-effects analyses to clarify which groups improved.

Minor Comments

1. Figures/Tables: Label axes in Figures 8–11 (e.g., "rSO₂ (%)" and "Time (min)"). Report exact p-values (e.g., "p = .012" vs. "p < .05") in Table 2 footnotes.

2. Compare findings with meta-analysis to explain null results (e.g., differences in exercise intensity/duration). When discussing ecological validity of the foraging task, explicitly link it to real-world cycling scenarios (e.g., attentional allocation on roads).

Language and Formatting

1. Standardize terms (e.g., "oxygen saturation" vs. "rSO₂").

2. Check the fonts in all the pictures, including their colors and types.

**Do you want your identity to be public for this peer review?** For information about this choice, including consent withdrawal, please see our Privacy Policy

Reviewer #1: No

Reviewer #2: No

---

## [Author Response · Author response to Decision Letter 1]

19 Sep 2025

PLOS One Revisions Response Document

Reviewer #1: General Comments

The present study examined the impact of ergometer cycling with visual foraging (EC + VF) on executive function (EF) and prefrontal cortex activation, in comparison with ergometer cycling (EC) and visual foraging (VF) alone. The research hypothesis was that the EC + VF condition would exhibit a more pronounced effect on EF and prefrontal cortex activation than either EC or VF alone. The research theme is interesting, and the findings may contribute to our understanding of the relationship between acute exercise and cognitive function. However, there are several serious concerns regarding the methods that the authors should address.

1) Sample Size Calculation via Power Analysis

The authors state that a power analysis using G*Power indicated an adequate sample size (Lines 122–127). However, the statistical model used in the study—a two-way repeated measures ANCOVA—is not well-supported by G*Power. In general, G*Power does not allow for accurate power analysis for repeated measures ANOVAs involving multiple within-subject factors, as these models include multiple random effects that are not easily distinguishable from error. In fact, G*Power version 3.1 does not include an option for computing within-within interactions for F-tests. The authors should provide a more detailed explanation of how the sample size was determined for the two-way repeated measures ANCOVA.

We agree, G*Power does not cater for all levels of design complexity – and consequently it cannot provide reliable estimates for complex designs such as ours and therefore may yield underestimates. We have acknowledged this potential compromise in the opening paragraph of the Methods, including acknowledgement of the fact that this required extensive – and intensive – participant contributions (lines 121-124):

“Sample size was estimated using G*Power 3.1 (35). Estimates were based on a published effect size of d = 0.52 for acute moderate intensity cycling on EF task performance (3), experimental power of 0.80, and a significance level of 0.05 for a repeated measures ANOVA (Condition [EC, VF, EC+VF] x Time [pre-, post-intervention), to yield a desired sample size of 27 participants, which is comparable to sample sizes for previous studies in which the effects of dual-tasking have been examined (17,36–38). However, G*Power does not derive estimates for interaction effects for repeated measures variables, and so this figure may be an underestimate. Nonetheless, because this was a pilot study – one that required multiple demanding laboratory visits from participants – we aimed to recruit this number, minimally.”

We hope this addresses your point suitably. Thank you.

2) Treatment of Confounders in fNIRS Data

In Lines 284–291, the features of the fNIRS device (INVOS 5100C) are described. However, the authors do not mention how potential confounders in fNIRS data, such as systemic physiological factors (e.g., respiration rate, heart rate), changes in scalp blood flow, and postural changes, were handled, even though it is well known that fNIRS data are vulnerable to these confounders. More details about fNIRS data processing should be provided. Additionally, it is unclear whether cerebral oxygenation was measured during the cognitive tasks. Typically, when investigating the association between cognitive functions and brain activation, fNIRS records oxyHb and deoxyHb during the cognitive tasks, not during exercise itself. Referring to Figure 10, the fNIRS data, represented as oxygen saturation (%), appear to have been measured during both exercise and VF conditions. The authors should clarify the validity and interpretation of fNIRS measurements in this study.

Thank you to the reviewers for this comment. We agree that fNIRS is susceptible to confounders. We have included the following information on lines 606-615 in the Discussion to acknowledge this:

“Our rationale for collecting fNIRS data during all three conditions was to examine how cortical oxygenation varies according to physical and cognitive demands, in isolation and combined. Therefore, our analysis focused on within-subject relative changes in rSO₂ to account for individual variability. A limitation is that the fNIRS device is susceptible to extracranial contamination, such as skull thickness and posture with no clear algorithm to correct measurements (90). However, we took steps to mitigate these confounding factors. These included maintaining consistent participant positioning by placing the fNIRS device after the participant was seated on the ergometer, instructing participants to minimize head movement and to perform the visual search task using eye movements rather than head movements. Additionally, we attempted to reduce external noise by turning off the room lights and closing the window blinds.”

3) Statistical Analyses

The authors used a two-way repeated measures ANCOVA with baseline EF performance as a covariate. However, this model may not be appropriate. Since both “Condition” (EC, VF, EC + VF) and “Time” (Pre, Post) are within-subject factors, the interaction between these two factors is sufficient to test for differential effects on EF. Including baseline EF performance as a covariate may be redundant. Moreover, because baseline scores are likely to be correlated with both pre- and post-test scores, their inclusion as covariates could introduce multicollinearity and bias the results. A repeated measures two-way ANOVA would likely be more appropriate.

We thank the reviewer for this comment. We understand the concern about potential redundancy; however, we have elaborated on this methodological decision in the Data Analysis subsection of the Methods (lines 359-365):

“Baseline EF performance was included as a covariate to control for individual differences and improve sensitivity in detecting intervention effects. Although ‘Time’ captures pre-post changes, adjusting for baseline reduces between-subject variability and potential bias. To address concerns about multicollinearity, we examined correlations between baseline and pre-intervention EF scores across conditions and tasks; all correlations ranged from .10 to .61, below the accepted multicollinearity threshold of r = .70. These results indicate that baseline EF and pre-intervention scores were not redundant, supporting the use of the ANCOVA model.”

The correlations ranged from low to moderate for the Flanker Task:

(EC+VF: r(21) - .27, VF: r(22) = .23, EC: r(22) = .51), WCST error count (EC+VF: r(20) = .25, VF: r(22) = .61, EC: r(22) = .23), and 2-Back correct matches (EC+VF: r(23) = .37, VF: r(25) = .31, EC: r(25) = .10.

4) Explanation of Gaze Fixation

Gaze fixation is a central variable in the current study. However, the description is insufficient; the manuscript only states that “Gaze data were automatically collected into the iMotions software” (Line 380). To ensure reproducibility using different devices or software, the authors should provide more detailed information about how gaze fixation was measured and defined.

Yes, in hindsight, our description was far too brief. We have added the following information on lines 370-377:

“Eye movements were recorded using Tobii Pro Glasses 2 at a sampling rate of 100 Hz. The glasses feature a wide-angle scene camera with a resolution of 1920 × 1080 pixels, capturing video at 25 frames per second with a 90-degree horizontal field of view (16:9 aspect ratio). The glasses were wirelessly connected to a PC running iMotions software (version 9.3). Fixations were automatically detected using iMotions’ built-in dispersion-based algorithm. Gaze behaviour was analysed as the number of fixations, which served as an index of visual attention. The mean number of fixations in the two visual foraging conditions was compared using paired samples t-tests.”

5) Other Issues (Typos and Errors)

There are numerous typographical and formatting errors throughout the manuscript. A thorough review and revision are recommended. Examples include:

-Line 154: “(see (41)” -> should be either “(see [41])” or “(see (41))”

-Line 216: “(hits/hits + errors) x 100” → should be “(hits / (hits + errors)) × 100”

-Line 341: “(see 3)” -> it is unclear what this refers to.

-Line 403: “F(2,42) = ηp2 = 0.14, p = 0.046” → the F-value appears to be missing.

-Line 437: “The upper panel of Figure 8 illustrates average rSO2 at the left optode, by Condition.” -> Figure 8 appears to display self-reported affect values, not rSO2.

-Line 445: “t(39) = 22.23.” -> Please verify the degrees of freedom; this value seems unusually high.

Thank you for flagging up these errors. We carefully re-read the entire manuscript for such errors, and consequently corrected not only the errors you identified, but also some additional ones.

Reviewer #2: This pilot study examines the effects of dual-task training (ergometer cycling + visual foraging) on executive function (EF), prefrontal cortex (PFC) activation, and gaze behavior. The topic is timely, and the experimental design is rigorous, incorporating fNIRS, eye-tracking, and validated EF tasks. While the findings contribute to the literature on exercise-cognition interactions, several issues require clarification to strengthen the manuscript.

Major Comments

1. The authors hypothesized that the combined condition (EC+VF) would yield greater EF improvements, yet the results did not support this (e.g., near-ceiling effects in EF tasks). This discrepancy should be discussed in more depth, addressing potential explanations such as task difficulty or individual differences. Consider higher-complexity EF tasks (e.g., 3-Back, Stroop) in future work to avoid such constraints.

We agree that potential ceiling effects and task simplicity may have limited our ability to detect possible effects across conditions, as we discuss on lines 590-599:

“Another limitation of this study may be the selected EF tasks. The complexity of EF tasks may influence the quality of the associated outcome measures (85), hence, employing more sophisticated tasks may yield larger effects (85–87). Our findings suggest that the EF tasks were relatively low in complexity, resulting in a near-ceiling effect in participants’ performance. For example, average pre-test scores for the 2-Back and Flanker tasks were 89% and 96%, respectively, leaving limited scope for improvement. Hence, higher-order EF task measures, such as those requiring reasoning and problem-solving (67), may be more appropriate. Utilizing more challenging tasks may minimize ceiling effects, enhance sensitivity, and engage multiple executive function domains, allowing for the detection of subtle changes that simpler tasks may not detect (88,89).”

2. The reduced cadence/energy in the EC+VF condition suggests dual-task costs, but the mechanistic explanation (e.g., resource competition vs. strategic trade-offs) is underdeveloped. The authors should: Cite dual-task frameworks (e.g., Kahneman’s Capacity Model) to contextualize findings. Discuss whether the observed PFC-gaze correlation in EC+VF reflects shared resources or compensatory effort.

Thank you for these insightful and useful comments. We have expanded the Discussion section to incorporate Kahneman’s model (1973), positing that cognitive and physical tasks compete for limited attentional resources. We have also addressed the possibility that the PFC-gaze correlation may reflect increased use of prefrontal responses due to compensatory effort (lines 579-587):

“The reductions in cadence and energy output may reflect dual-task costs, consistent with Kahneman’s Capacity Model of Attention (1973), which suggests that cognitive and physical tasks compete for limited attentional resources. Therefore, the demands of the EC+VF task may have exceeded available attentional capacity, leading to lower cadence levels as participants may have prioritized the VF over EC. Additionally, the correlation between PFC activation and gaze fixations in the EC+VF condition may indicate demands on a shared neural resource or compensatory efforts to maintain performance. More temporally or spatially precise neurophysiological data, such as those obtained via EEG and fMRI data, respectively, would help to elucidate such mechanisms.”

3. Exercise Intensity Control: Cadence varied significantly between EC and EC+VF (Fig. 11). Was intensity matched via HR or RPE? If not, this confounds comparisons. fNIRS Data: Specify whether rSO₂ changes were normalized to baseline. Correlations between rSO₂ and behavioral outcomes (beyond gaze) would bolster claims about neural efficiency.

These are good points. We have clarified our methods as follows:

Standardisation of Exercise Intensity:

“Participants were instructed before each condition to maintain a target cadence of 60-70 rpm. Further, exercise intensity was standardized across participants using individualised V˙O2max testing. As each participant exercised at a fixed percentage of their individual V˙O2max (60%), we ensured a consistent relative intensity across the conditions. As the resistance remained constant across the exercise conditions and only cadence fluctuated slightly within the instructed range, the relative exercise intensity was controlled.” (Lines 231-236)

Normalisation of fNIRS Data:

“rSO2 values were not normalised to baseline but were analyzed as absolute rSO2 values. This enabled direct comparison of rSO2 levels across conditions without transformation. (Lines 284-286).

Correlations between rSO₂ and behavioural outcomes:

We appreciate the suggestion to examine associations between cerebral oxygenation and behavioural performance. We ran an exploratory analysis but did not find any significant correlations, all p’s > .05. This is noted in the Results section:

“An exploratory analysis was conducted to examine potential relationships between rSO₂ and behavioral outcomes, but none emerged (all p’s > .05).” (Lines 458-459)

The findings of the correlations are presented in the table below:

Correlations between rSO₂ and behavioural outcomes

rSO₂ X Pre-to-Post Flanker EC r(25) = .31 p = .112

rSO₂ X Pre-to-Post Flanker VF r(25) = .00, p = .998

rSO₂ X Pre-to-Post Flanker EC+VF r(24) = .23, p = .250

rSO₂ X Pre-to-Post WCST EC r(25) = .18, p = .361

rSO₂ X Pre-to-Post WCST VF r(25) = .20, p = .321

rSO₂ X Pre-to-Post WCST EC+VF r(24) = .15, p = .451

rSO₂ X Pre-to-Post 2-Back EC r(25) = .05, p = .801

rSO₂ X Pre-to-Post 2-Back VF r(25) = .07, p = .738

rSO₂ X Pre-to-Post 2-Back EC+VF r(23) = .02, p = .914

4. The Condition × Time × Covariate interaction for the Flanker task (p = 0.038) is a key finding but lacks post-hoc detail. Include effect sizes (e.g., η²) and simple-effects analyses to clarify which groups improved.

Thank you for pointing out this oversight. Given that it is not possible to stratify data according to a continuous covariate, we performed a median split according to baseline scores, to examine pre-test-post-test differences in Flanker task scores. We have reported this as follows on lines 400-408:

“However, there was a significant interaction between condition, time and baseline EF task scores (the covariate), F(2,42) = 3.54, ηp2 = .14, p = .038. To evaluate this interaction, we conducted simple effects analyses using a median split on the baseline scores. Among participants with better performance on the Flanker task at baseline (Median Scores = 17 ms), significant improvement was shown after the EC condition in the Flanker task performance, t(10) = 3.16, p = .01, d = 0.95, but not the VF or EC+VF conditions (p > .05). In contrast, for participants with poorer performance (i.e., lower than the median), no significant change was observed in any of the conditions. These results suggest that EC was particularly beneficial for individuals with higher baseline Flanker Scores.”

Minor Comments

1. Figures/Tables: Label axes in Figures 8–11 (e.g., "rSO₂ (%)" and "Time (min)"). Report exact p-values (e.g., "p = .012" vs. "p < .05") in Table 2 footnotes.

Thank you for these suggestions, too. We have updated figure axes, but we decided not to report p values in footnotes in Table 2 for three reasons: (1) p values associated with all comparisons are already reported in the body of the manuscript, (2) the ta

---

## [Decision Letter · Decision Letter 1]

12 Oct 2025

Dear Dr. Bishop,

Thank you for submitting your manuscript to PLOS ONE. After careful consideration, we feel that it has merit but does not fully meet PLOS ONE’s publication criteria as it currently stands. Therefore, we invite you to submit a revised version of the manuscript that addresses the points raised during the review process.

We look forward to receiving your revised manuscript.

Kind regards,

Takao Yamasaki

Academic Editor

PLOS ONE

Journal Requirements:

Additional Editor Comments:

The authors need to make minor revisions in response to reviewer 1's comments.

Reviewers' comments:

Reviewer's Responses to Questions

**Comments to the Author**

Reviewer #1: (No Response)

Reviewer #2: All comments have been addressed

2. Is the manuscript technically sound, and do the data support the conclusions?

Reviewer #1: Yes

Reviewer #2: Yes

3. Has the statistical analysis been performed appropriately and rigorously?

Reviewer #1: Yes

Reviewer #2: Yes

4. Have the authors made all data underlying the findings in their manuscript fully available?

Reviewer #1: Yes

Reviewer #2: Yes

5. Is the manuscript presented in an intelligible fashion and written in standard English?

Reviewer #1: Yes

Reviewer #2: Yes

Reviewer #1: The authors have addressed most of the comments raised previously. Although I do not fully agree with all of the responses and additional explanations, the revised manuscript is generally convincing. However, the issue of the power analysis still remains.

First, I agree that a sample size of 27 is adequate for the current study. Nevertheless, as noted in my previous comment, G*Power cannot directly perform a power analysis for a two-way repeated measures ANOVA model. Therefore, I assume that the authors conducted the analysis using either “ANOVA: Repeated measures, within factors,” “ANOVA: Repeated measures, within–between interaction,” or another approximation. The current explanation does not provide sufficient detail for readers to reproduce the analysis. I encourage the authors to clearly specify the exact procedure they used in G*Power.

Furthermore, the reported effect size is inappropriate. For repeated measures ANOVA, the conventional indices are partial eta-squared (η²p) or f, not Cohen’s d as stated in the manuscript. It would strengthen the manuscript if the authors could report the appropriate effect size (η²p or f) used in the power analysis.

Reviewer #2: I thank the authors for their thoughtful responses and great efforts to address the comments. I have no further comments for the authors to address at this time.

**Do you want your identity to be public for this peer review?** For information about this choice, including consent withdrawal, please see our Privacy Policy

Reviewer #1: No

Reviewer #2: No

---

## [Author Response · Author response to Decision Letter 2]

25 Oct 2025

Reviewer #1: The authors have addressed most of the comments raised previously. Although I do not fully agree with all of the responses and additional explanations, the revised manuscript is generally convincing. However, the issue of the power analysis still remains.

First, I agree that a sample size of 27 is adequate for the current study. Nevertheless, as noted in my previous comment, G*Power cannot directly perform a power analysis for a two-way repeated measures ANOVA model. Therefore, I assume that the authors conducted the analysis using either “ANOVA: Repeated measures, within factors,” “ANOVA: Repeated measures, within–between interaction,” or another approximation. The current explanation does not provide sufficient detail for readers to reproduce the analysis. I encourage the authors to clearly specify the exact procedure they used in G*Power.

Thank you for your helpful comment regarding the sample size estimation – and please accept our apologies for not fully addressing your comment first time around. We have now explicitly explained the power analysis procedure in the revised manuscript. This information has been added to the Methods section (Lines 116-130) to increase transparency and reproducibility:

“Sample size was estimated using G*Power 3.1 (35). Estimates were based on a published effect size of d = 0.52 for acute moderate intensity cycling on EF task performance (3), experimental power of 0.80, and a significance level of 0.05 for a repeated measures ANOVA (Condition [EC, VF, EC+VF] x Time [pre-, post-intervention), to yield a desired sample size of 17 participants. However, G*Power does not derive estimates for interaction effects for mixed-design repeated measures ANOVAs, this figure may represent an underestimation. Further, because this was a pilot study – one that required multiple demanding laboratory visits from participants leading to potential participant attrition - a total of 27 participants were recruited. This sample size is also comparable to those used in previous studies examining the effects of dual-tasking (17,36–38).

We adopted the ‘ANOVA: Repeated measures, within factors’ G*Power analysis protocol to approximate the sample size for a within-subjects design. The parameters were as follows: an effect size of Cohen’s f = 0.26 (converted from Cohen’s d = 0.52), α = 0.05, power = 0.80, one group, and six repeated measurements (derived from 3 conditions x 2 time points). This analysis yielded a required sample size of 17 participants. However, to account for potential underestimation, participant attrition, and comparability with previous studies, we aimed to recruit 27 participants.”

---

## [Decision Letter · Decision Letter 2]

29 Oct 2025

The Effect of Ergometer Cycling and Visual Foraging on Brain Function: A Pilot Study

PONE-D-25-20437R2

Dear Dr. Bishop,

We’re pleased to inform you that your manuscript has been judged scientifically suitable for publication and will be formally accepted for publication once it meets all outstanding technical requirements.

Kind regards,

Takao Yamasaki

Academic Editor

PLOS ONE

Additional Editor Comments (optional):

Reviewers' comments:

Reviewer's Responses to Questions

**Comments to the Author**

Reviewer #1: All comments have been addressed

2. Is the manuscript technically sound, and do the data support the conclusions?

Reviewer #1: Yes

3. Has the statistical analysis been performed appropriately and rigorously?

Reviewer #1: Yes

4. Have the authors made all data underlying the findings in their manuscript fully available?

Reviewer #1: Yes

5. Is the manuscript presented in an intelligible fashion and written in standard English?

Reviewer #1: Yes

Reviewer #1: (No Response)

**Do you want your identity to be public for this peer review?** For information about this choice, including consent withdrawal, please see our Privacy Policy

Reviewer #1: **Yes: ** Shinji Takahashi

---

## [Editor Report · Acceptance letter]

PONE-D-25-20437R2

PLOS ONE

Dear Dr. Bishop,

I'm pleased to inform you that your manuscript has been deemed suitable for publication in PLOS ONE. Congratulations! Your manuscript is now being handed over to our production team.

Kind regards,

on behalf of

Dr. Takao Yamasaki

Academic Editor

PLOS ONE